# Can Retriever-Augmented Language Models Reason? The Blame Game Between the Retriever and the Language Model

**Parishad BehnamGhader**[1]    **Santiago Miret**[3]    **Siva Reddy**[1,2]

[1]McGill University / Mila    [2]Facebook CIFAR AI Chair    [3]Intel Labs
{parishad.behnamghader, siva.reddy}@mila.quebec
santiago.miret@intel.com

## Abstract

Augmenting pretrained language models with retrievers has shown promise in effectively solving common NLP problems, such as language modeling and question answering. In this paper, we evaluate the strengths and weaknesses of popular retriever-augmented language models, namely $k$NN-LM, REALM, DPR + FiD, Contriever + ATLAS, and Contriever + Flan-T5, in reasoning over retrieved statements across different tasks. Our findings indicate that the simple similarity metric employed by retrievers is insufficient for retrieving all the necessary statements for reasoning. Additionally, the language models do not exhibit strong reasoning even when provided with only the required statements. Furthermore, when combined with imperfect retrievers, the performance of the language models becomes even worse, e.g., Flan-T5's performance drops by 28.6% when retrieving 5 statements using Contriever. While larger language models improve performance, there is still a substantial room for enhancement. Our further analysis indicates that multihop retrieve-and-read is promising for large language models like GPT-3.5, but does not generalize to other language models like Flan-T5-xxl.[1]

## 1 Introduction

Parametric language models, such as decoder-only transformers (e.g. GPT), transformer encoder models (e.g. BERT), and encoder-decoder transformers (e.g. T5), encode all necessary knowledge to solve a given task in their parameters and have demonstrated exceptional performance on many natural language tasks (Vaswani et al., 2017; Radford et al., 2018; Devlin et al., 2019; Raffel et al., 2020). Non-parametric models improve these models further by augmenting them with knowledge retrievers (Guu et al., 2020; Izacard and Grave, 2021; Izacard et al., 2022b) or memory components (Khandelwal et al.,

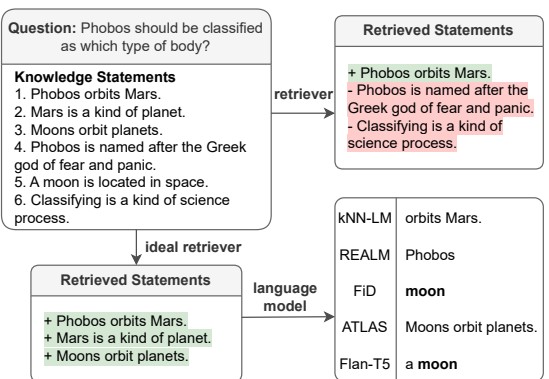

Figure 1: **Example of retriever and language model failures when reasoning is needed.** The correct and incorrect retrievals are highlighted in green and red, respectively. This example demonstrates that the retrievers' similarity metric is insufficient for retrieving required statements, and language models cannot perform reasoning over the retrieved statements perfectly.

2020; Verga et al., 2021; Zhong et al., 2022). This augmentation helps them acquire new knowledge on-the-fly from external sources rather than relying solely on the implicit knowledge encoded in the model's parameters, thereby making them more robust to domain shifts (Izacard et al., 2022b). Furthermore, the retrieved knowledge can provide insight into what knowledge the model is using.

While the capabilities of parametric language models have been extensively studied in the literature (Wei et al., 2022; Zelikman et al., 2022), there is no thorough study of the limitations of non-parametric models. For instance, Mallen et al., 2023 examine the performance of non-parametric memories when encountering less popular knowledge. In contrast, our work takes a systematic approach to study the limitations of retriever-augmented language models for reasoning over retrieved information. As depicted in Figure 1, these models often fail to solve the tasks that require sequential logical reasoning, such as taxonomic chaining and combining the details.

---

[1]The code is available at https://github.com/McGill-NLP/retriever-lm-reasoning.

In this study, we demonstrate how retriever-augmented language models struggle with logical reasoning when the task involves reasoning over multiple statements. We evaluate these models in language modeling (LM) and question answering (QA) tasks using different variations of EntailmentBank (Dalvi et al., 2021) and StrategyQA (Geva et al., 2021) datasets, where we have control over the provided supporting statements and reasoning skills. Notably, these datasets do not explicitly indicate the reasoning path within the question itself, making the retrieval process more challenging. In other words, knowledge statements and queries may not have surface-level lexical similarity. For instance, the question in Figure 1 has no lexical similarity with required statements (2) and (3), and without a strong reasoning component, models would struggle to retrieve and reason upon such statements.

Concretely, we analyze the performance of $k$NN-LM, REALM, DPR + FiD, Contriever + ATLAS, and Contriever + Flan-T5. As illustrated in Figure 1, these models exhibit shortcomings rooted in both parts of their design: 1) Retrievers struggle to select all the necessary statements for reasoning when using the similarity between query and knowledge statements (§4.3.1); 2) Language models are imperfect reasoners even when provided with a perfect retriever that retrieves all the essential information (§4.3.2); 3) Moreover, the performance of language models deteriorates further if the retriever is imperfect, a closer setting to reality (§4.3.3); 4) Additionally, experimental results indicate that while larger language models yield improvements, even the largest models we studied are imperfect reasoners (§4.3.4); 5) Finally, we observe that employing multihop retrieve-and-read enhances GPT-3.5's performance in reasoning tasks, but this improvement does not extend to other models such as Flan-T5-xxl (§4.3.5).

## 2 Background

There has been a growing interest in investigating the reasoning abilities of parametric language models (Wei et al., 2022; Chung et al., 2022). In contrast, our work focuses on the reasoning abilities of retriever-augmented language models. This paper further studies the contributions of each of the models' components.

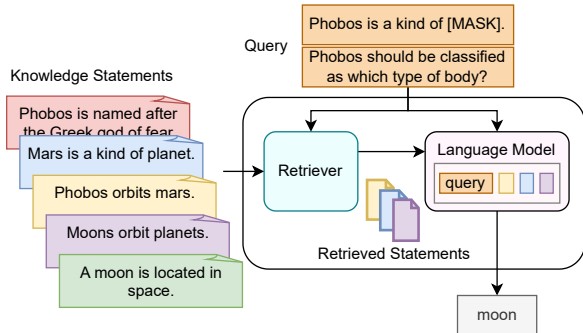

Figure 2: **The architecture of retrieve-then-read retriever-augmented language models.** The language model predicts the answer using the query and the retriever's selected statements.

### 2.1 Retriever-Augmented Language Models

Major work in retriever-augmented language models focused on improving downstream performance such as question answering and language modeling using *retrieve-then-read* paradigm (Guu et al., 2020; Izacard and Grave, 2021; Izacard et al., 2022b; Khandelwal et al., 2020). Figure 2 illustrates a generic retrieve-then-read architecture. Some of the popular models that we study are $k$NN-LM, Retrieval-Augmented Language Model (REALM), Fusion-in-Decoder (FiD), and ATLAS (Khandelwal et al., 2020; Guu et al., 2020; Izacard and Grave, 2021; Izacard et al., 2022b). Each of these leverages retrievers and language models in a unique manner.

### 2.2 Role of Retriever

The retriever's role is to retrieve relevant statements for a given query which are then used by the language model. Dense Passage Retriever (**DPR**), **Contriever**, and REALM's retriever fetch the most similar statements based on a similarity metric (i.e., dense inner product) between query and statements' representations using two independently trained or one pre-trained BERT-based encoder (Karpukhin et al., 2020; Izacard et al., 2022a; Guu et al., 2020). In contrast, $k$NN-LM adopts an L2 similarity metric between the representation of the query and partial token sequences (instead of full knowledge statements) to select the most relevant sequences of tokens (Khandelwal et al., 2020). While retrievers typically select statements from a large common corpus in the literature, as depicted in Figure 2, we provide a data-specific set of statements for each query to control the supporting information.

## 2.3 Role of Language Model (Reader)

The role of the language model is to make use of the retrieved knowledge to generate relevant text for the given input query. **$k$NN-LM**, a decoder-only Transformer, computes the distribution over the next token during generation by interpolating between the transformer's next token distribution and the next token from the nearest neighbor memory (Khandelwal et al., 2020). On the other hand, **REALM** is a masked language model backed by a BERT-based reader that extracts the most promising span from one of the statements as the answer (Guu et al., 2020).

**FiD** and **ATLAS** are both sequence-to-sequence T5-based neural networks (Izacard and Grave, 2021; Izacard et al., 2022b). ATLAS is specifically designed for jointly finetuning the language model and the retriever, employing various pretext tasks with limited training examples. In these models, the encoder encodes the query and each retrieved statement individually, and the decoder attends to these representations to solve the downstream task.

While **Flan-T5** was not specifically built for retriever-based language modeling, since it is an instruction-tuned model, it can be combined with any retriever to complete downstream tasks using the retrieved information (Chung et al., 2022).

## 2.4 Multihop Retrieve-and-Read

In addition to the previously mentioned widely used *retrieve-then-read* models, multihop retrieve-and-read iteratively utilizes textual or dense search queries for a fixed predefined or variable number of iterations (Xiong et al., 2021; Qi et al., 2021; Khot et al., 2020; Khattab et al., 2022). For instance, Demonstrate-Search-Predict (**DSP**) can be used in multihop question answering. In each iteration, DSP employs large language models to generate a query by decomposing a complex question into smaller subproblems and summarizes information from retrieved knowledge (Khattab et al., 2022).

## 3 Problem Definition

In the main experiments, we provide the models with a complete set of knowledge statements denoted as $S = \{s_1, s_2, \ldots, s_m\}$ for each sample. In some cases, only a subset of these statements is essential for predicting the answer, referred to as *gold statements*. The primary objective of the models is to 1) retrieve a set of statements $S_r = \{r_1, r_2, \ldots, r_k\} \subseteq S$ which find necessary

| Model | # Params | Model | # Params |
|---|---|---|---|
| Language Models | | | |
| REALM | ~270M | FiD | ~220M |
| $k$NN-LM | ~250M | ATLAS | ~250M |
| Flan-T5-base | ~250M | | |
| Model Size and Multihop Retrieve-and-Read Analysis | | | |
| Flan-T5-small | ~80M | Flan-T5-xl | ~3B |
| Flan-T5-base | ~250M | Flan-T5-xxl | ~11B |
| Flan-T5-large | ~780M | GPT-3.5 | ~175B |

Table 1: **The number of model parameters.** We control for model size to circumvent the role of size in reasoning abilities. Moreover, we evaluate the impact of model size and multihop retrieve-and-read using larger models.

and 2) effectively solve the target task through reasoning over the retrieved knowledge $S_r$. A visualization of the task is illustrated in Figure 2. We control for model size wherever possible for comparable results among different models. We also study the effect of scaling the model size. Table 1 presents the number of parameters of the studied models. Appendix A contains additional implementation details. We analyze the performance of these models on two tasks, Language Modeling (LM) and Question Answering (QA), using datasets (Section 4.1) that are specifically curated for reasoning.

**Language Modeling (LM).** In the language modeling setup, we evaluate the effectiveness of the retriever-augmented language models in a *target ranking* task. In this task, the model should assign a higher likelihood to the correct sentence compared to (up to) four alternative similar but incorrect sentences. For instance, in the example illustrated in Figure 2, the models should rank the sentence *"Phobos is a kind of moon."* higher than an alternative sentence such as *"Phobos is a kind of planet."* This task allows comparing masked language models like REALM with autoregressive models. We explain this task in more detail in Appendix C.1 and present its experimental results in the Appendix D.

**Question Answering (QA).** In the question answering setup, the model should answer a question given retrieved statements, as illustrated in Figure 2. Except for $k$NN-LM, all the other models are already exposed to question answering tasks during their training.

## 4 Experimental Setting

### 4.1 Datasets

We assess the performance of the models using the following reasoning datasets. These datasets enable us to evaluate the models' reasoning abilities and the retrievers' performance in LM and QA tasks while controlling for the available supporting information.

**EntailmentBank** (**EB**, Dalvi et al., 2021) consists of an input question or a hypothesis statement that can be inferred only through multi-step reasoning on the provided statements. For the QA task, we use the question format of the input, and for the LM task (i.e., target ranking), we use the hypothesis statement and alternative statements obtained by replacing the target entity with other alternative entities (see Appendix B). As knowledge statements, we use the provided gold and distracting statements in the dataset. The distracting statements mostly contain entities or relations mentioned in the input query which might make them deemed relevant but in actuality they are irrelevant. For instance, for the question *"Which rock type is most useful in studying the history of living organisms?"*, sentecnes *"Nearly all fossils are found in sedimentary rock."* and *"Limestone is a kind of sedimentary rock."* are relevant, and sentences *"Rock means stone."* and *"Organisms can be preserved in sedimentary rock."* are distracting and irrelevant. EB consists of multiple splits but we use the EB-2 split since its knowledge store contains both required and distracting statements. We call EB-2 split as EB-Hard in this paper. For each input, there will be up to 25 statements, of which up to 12 statements are required to answer correctly. EB-1 is similar to EB-2 but without any distracting statements. EB also has EB-3 data which includes 25 relevant and irrelevant statements sampled from a large corpus, but we find the trends to be similar to EB-2 (see Appendix D).

**StrategyQA** (Geva et al., 2021) contains *yes* or *no* questions accompanied by up to 5 supporting statements from Wikipedia. To evaluate the models in the language modeling setting, we convert each question and answer into a declarative statement, while also generating incorrect statements using alternative entities. The detailed dataset preparation process is explained in

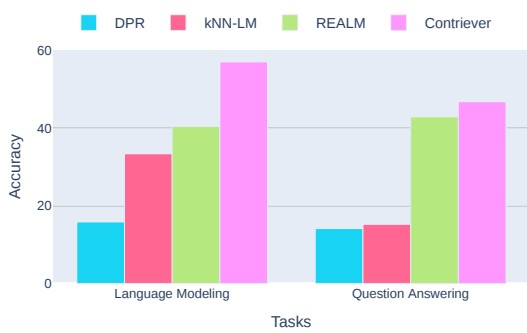

Figure 3: **Retrievers' accuracy on EB-Hard test set in LM and QA tasks.** Results show that retrievers do not select required statements properly, as the best retriever, Contriever, achieves only a 47% accuracy in QA task.

Appendix B.

### 4.2 Evaluation Metrics

In the QA setting, we use the token overlap F1 score between the predicted answer and the gold answer. For the LM task, we evaluate the accuracy of the models in the *target ranking* problem.

For the retriever, we use accuracy and recall score to indicate the overlap between retrieved statements and the ground-truth statements which are essential to obtain the answer.

### 4.3 Evaluation and Discussion

In this section, we analyze the limitations of both retrievers and language models in reasoning tasks.

#### 4.3.1 The Shortcomings of Retrievers

Current retrievers select $k$ statements based on a relevance score, such as inner product or L2 distance between the query's and statements' (or spans of statements) representations (Guu et al., 2020; Karpukhin et al., 2020; Izacard et al., 2022a; Khandelwal et al., 2020). However, this approach is insufficient for reasoning. Consider the question *"Phobos should be classified as which type of body?"* in Figure 1. The retriever may select similar statements to the query, such as (1)*"Phobos orbits Mars."* and (4)*"Phobos is named after . . . "*, but none of them contains the answer "moon." Instead, combining statement (1) with missed statements (2)*"Mars is a kind of planet."* and (3)*"Moons orbit planets."* would provide the answer to the question. But statements (2) and (3) are the ones with the least similarity.

We validate our hypothesis that the similarity-based metric for retrieval is insufficient for reasoning using the EB-Hard dataset. Since the EB-Hard

| Model | Query | Statements | Prediction |
|---|---|---|---|
| DPR + FiD | In a zoo located in a warm region, what should be included in the polar bear exhibit? | + If an animal lives a certain environment then that animal usually requires that kind of environment. 
 - Polar bears live in **cold environments**. | warm |
| Contriever + ATLAS | What keeps the Moon orbiting Earth? | + Moons orbit planets. 
 - **Gravity** causes orbits. | elliptical |
| kNN-LM | The robot will weigh less on mars than earth but will have the same [MASK]. 
 Targets: *mass* vs *mars* | + As the force of gravity decreases, the weight of the object will decrease. 
 - The gravitational force of a planet does not change the **mass** of an object on that planet or celestial body. | mars |

Table 2: **Some examples of models' failures rooted in the retriever.** One of the correctly retrieved statements and the one that had to be retrieved in order for the model to solve the task correctly are highlighted in green and red, respectively. The sequence of tokens leading to the true answer is marked in **bold**. Results show that current retrievers struggle to retrieve required statements for solving the tasks.

dataset provides us with both gold and distracting statements to answer a question or infer a hypothesis, we evaluate the accuracy of different retrievers when we know $k$, i.e., the exact number of required statements to retrieve for a given input. We present the results in Figure 3. The best-performing retriever, Contriever, achieves only 57% and 47% accuracy in the LM and QA tasks, respectively. DPR, the widely used retriever with BERT-based dual encoder trained on Natural Questions (Kwiatkowski et al., 2019), is the worst with an accuracy of only around 15%. REALM's retriever, which is a DPR trained on self-supervised large-scale salient span masking on Wikipedia and fine-tuned on Natural Questions improves the performance by a large margin but still has only 40% accuracy. kNN-LM's retriever is trained on Wikitext-103 (Merity et al., 2017) in an auto-regressive fashion to select statements that help predict the next word of the input. Due to this, its performance drops 16 points when evaluated for the QA task compared to the LM task, as the QA task is out-of-distribution for kNN-LM.

Figure 4 further illustrates the recall score of the retrieved statements for varying $k$. While all retrievers reach a recall of 100% at 25 (i.e., the maximum number of knowledge statements for each sample in EB-Hard), kNN-LM still struggles due to the way it represents the facts in its memory component. In fact, each statement $s = w_1 w_2 \ldots w_n$ is stored as $n - 1$ key-value pairs in memory. For instance, the key $w_1$ is paired with value $w_2$ (i.e., next token), similarly $w_1 w_2$ with value $w_3$, and so on. The retriever computes the similarity between the input query and all the keys and selects the top $k$ similar keys (i.e., sequences of tokens) among all. When allowing kNN-LM to retrieve even 100 keys,

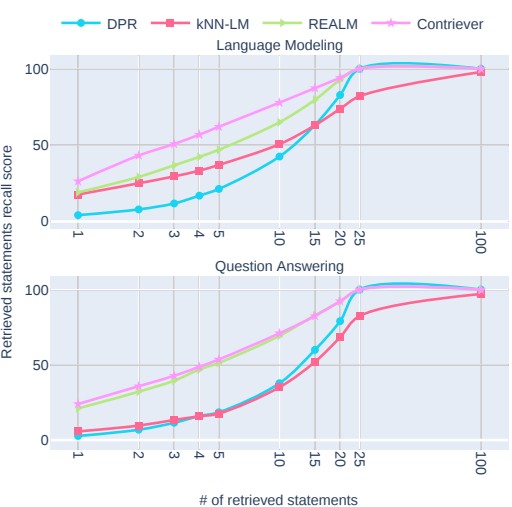

Figure 4: **Retreivers' recall score on EB-Hard test set in LM and QA based on the number of retrieved statements (k).** Contriever is shown to be superior among the studied retrievers. Results further indicate that kNN-LM does not cover 100% of the gold statements when $k = 100$ (kNN-LM's recall is $\approx 97\%$).

it still fails to cover 100% of the gold statements.

Failures of retrievers illustrated in Table 2 demonstrate that relying solely on query-statement similarity for retrieval may lead to overlooking important statements that are dissimilar to the query but contain information essential for reasoning.

### 4.3.2 The Shortcomings of Language Models

The language model has to reason upon the retrieved statements to answer a given input. In this section, we assume access to a perfect retriever.

To estimate the upper-bound performance of each language model, we use the final single statement hypothesis available in EB datasets. This statement can be entailed using an entailment tree

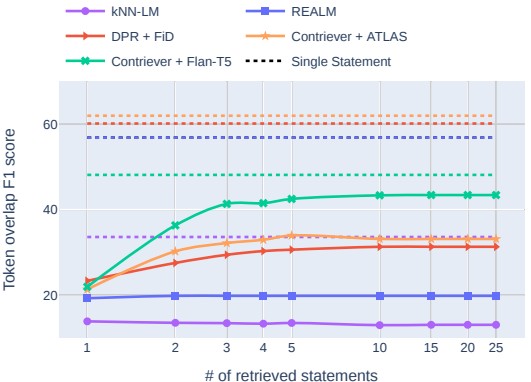

Figure 5: **Token overlap F1 score of language models on EB-Hard QA test set.** The dotted and solid lines refer to experiments given the single oracle statement and gold statements (when reasoning is required), respectively. Results illustrate that language models perform worse when answering the question requires reasoning.

and is sufficient to answer the given input. For the example in Figure 2, this statement is "*Phobos is a kind of moon*", which we refer to as the *oracle statement*. The oracle statement is provided to the model as the retrieved statement to answer the question "*Phobos should be classified as which type of body?*" (or, for the LM task, to rank the sentence "*Phobos is a kind of moon*" higher than others which is trivial).[2]

Figure 5 shows the results of QA task in dotted lines. The models ATLAS, FID, and REALM perform the best on this task, as these models are mainly trained to perform QA.

Table 3 shows some outputs of various models. These oracle scores would give an estimate of the upper-bound performance of models when the task does not require reasoning over multiple statements. Surprisingly, Flan-T5 performs worse than expected as the model struggles to limit itself to the provided knowledge and relies on its parametric memory, a common challenge faced by large language models in non-parametric setting (Mallen et al., 2023). This claim is backed by our further analysis on 50 randomly sampled (from among 340 samples) Flan-T5 responses given the oracle statement. The statistics shown in Figure 6 show that 18% of the Flan-T5 responses are not grounded in the oracle statement, even though the oracle statement was relevant to the question and answer. Additionally, 32% of the responses yield

---

[2]For the LM task, the models achieve high performance since the resulting statement is the same as the target statement among the alternatives.

| **Question**: The planets revolve in a counterclockwise direction. The cause of the revolution is mostly due to which force? 
 **Statement**: Gravity causes planets in the solar system to orbit the sun. 
 **Expected answer**: gravitational | | | |
|---|---|---|---|
| Model: | ATLAS 
 Flan-T5 | Response: | gravity 
 gravity |
| **Question**: When compared to the Sun, red dwarf stars are 
 **Statement**: Red dwarf stars are cooler than the sun. 
 **Expected answer**: cooler | | | |
| Model: | ATLAS 
 Flan-T5 | Response: | cooler than the sun. 
 a lot smaller |

Table 3: **The predictions of models when the oracle statement that contains the answer is provided.** F1 metric unnecessarily penalizes ATLAS here, although its predictions are correct. Flan-T5's response to the second question is not grounded to the oracle statement and the model is only using its internal knowledge to answer the question.

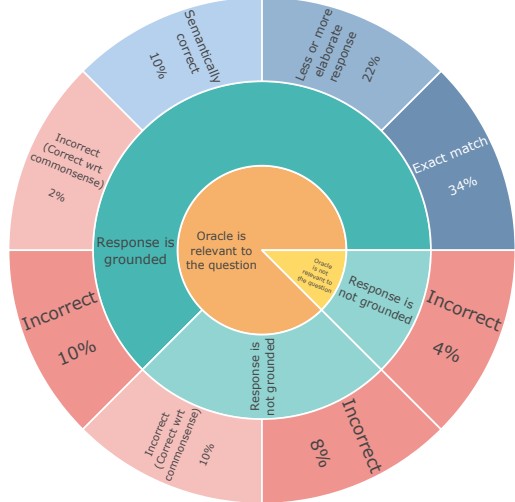

Figure 6: **Human analysis of Flan-T5's responses given the oracle statement.** As demonstrated in Table 3, 18% of the Flan-T5 responses are not grounded in the oracle statement and 32% of the responses yield a low F1 score even though they are correct.

a low F1 score even though they are correct (due to elaborateness or wording mismatch). The scores are lower than expected since the F1 evaluation metric penalizes when there is a word mismatch between true and predicted answer even if the predicted answer is semantically the same (Kamalloo et al., 2023; Chiesurin et al., 2023; Adlakha et al., 2023). Some other examples of Flan-T5's responses are also demonstrated in Table 10.

Additionally, we experiment with a setting

where the models have to reason on multiple retrieved gold statements. Figure 5 shows the results in solid curves. As we provide the models with an increasing number of gold statements that are essential to answer the input, the performance goes up for Flan-T5, FiD, and ATLAS. However, we do not observe any change in REALM and $k$NN-LM. Although REALM is a QA model, its limitations stem from how it makes use of retrieved documents. Instead of reasoning on all documents jointly, like ATLAS, FiD, and Flan-T5, it reasons on each document separately. In pretraining phase, the model marginalizes the score of each span over all documents. In the inference phase, the model picks the span with the highest retriever and reader score. In our setting, this almost always ends up selecting a span from the first few statements, as the statements are in the order of importance (see Figure 1). We present additional examples in Table 4. $k$NN-LM is not designed to perform QA, which is why it performs worse. Moreover, in our LM experiments, we find that it also underperforms other models, so it is unclear when kNN-LM style decoding is preferred (see Appendix D.1).

When we contrast the results of reasoning over multiple retrieved gold statements (solid curves in Figure 5) with only reasoning on the oracle statement (dotted), the performance of all models is much lower than the oracle performance even after providing all the essential gold statements. Interestingly, although Flan-T5's oracle performance is lower than ATLAS, FID, and REALM, it outperforms them when reasoning over multiple statements, indicating that it is a better reasoner. We conjecture that Flan-T5's multi-task training which also includes reasoning tasks like chain-of-thought reasoning, makes it a better reasoner than others.

Furthermore, we investigate the impact of additional distracting information on the performance of language models in the QA task in Figure 7. Flan-T5's performance drops by 8.7% in the presence of distracting statements indicating that irrelevant information hurts the performance of language models. Although ATLAS looks relatively robust, its performance is lower than Flan-T5, so we cannot draw a definitive conclusion.

On the LM task, we observe similar trends to the QA task and present the results and analysis in Appendix D.1.

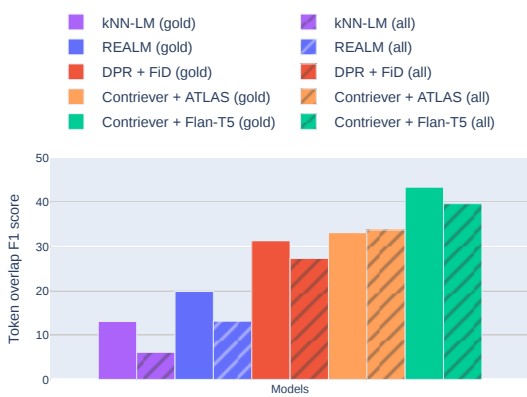

Figure 7: **The negative impact of additional distracting information on language models' performance in the QA task.** The solid bars and the bars with patterns refer to the experiments with all the gold statements and all gold and distracting statements, respectively. It can be observed that providing language models with distracting statements hurts their performance.

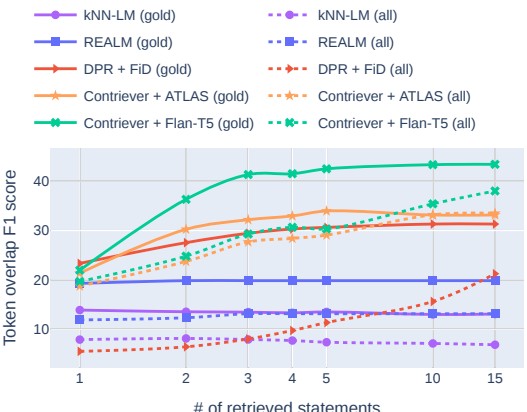

Figure 8: **Performance of language models on QA EB-Hard test set coupled with imperfect retrievers.** Coupling language models with imperfect retrievers (i.e., when a retriever fetches a distracting statement) deteriorates the overall performance.

### 4.3.3 The Blame Game: The Impact of Combining Imperfect Retrievers and Language Models

This section explores how the combination of language models and imperfect retrievers exacerbates the performance of retriever-then-read models, a setting closer to reality. An incorrect final answer could be blamed on the retriever's inability to fetch relevant statements or the failure of the language model to reason on the retrieved statements.

Additionally, Figure 8 illustrates the performance of language models on QA EB-Hard dataset, when coupled with either ideal retrievers (given only gold statements) or imperfect retrievers (given

| Model | Query | Retrieved statements | Prediction |
|---|---|---|---|
| Flan-T5 | What allows two students standing ten feet apart to hear each other talk? | + Talking is when a human produces sound to communicate.
+ Sound can travel through air by **vibrating air**. | a microphone |
| REALM | Andy lives in the southern hemisphere. What season does he most likely experience in August? | + Andy lives in southern hemisphere.
+ August is during the **winter** in the southern hemisphere. | in southern hemisphere |

Table 4: **Some examples of models' failures rooted in the language model.** In each example, two correct retrieved statements are illustrated. The true answer is marked in **bold**. We observe that 1) Flan-T5 does not limit itself to the provided knowledge, and 2) REALM extracts the response from the first retrieved statement, disregarding other retrieved statements.

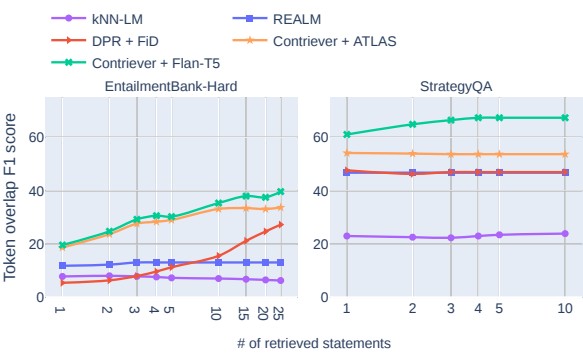

Figure 9: **Performance of the retrieval-augmented models in QA on test sets based on the number of retrieved statements.** The results demonstrate that although Contriever + Flan-T5 and Contriever + ATLAS are superior, the studied models perform poorly at reasoning when answering questions.

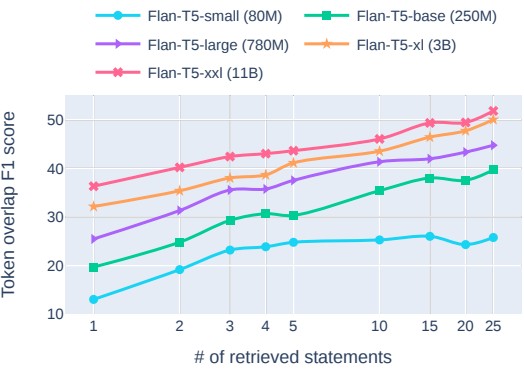

Figure 10: **Token overlap F1 score of Flan-T5 variations on EB-Hard QA test set.** The results reflect the impressive impact of size on the performance of the models in the tasks where reasoning is required. We use Contriever as the retriever in all experiments.

gold and distracting information). The results reveal a significant performance gap, as Flan-T5's performance in the QA experiment drops by 28.6% when retrieving 5 statements using Contriever. We further report the influence of imperfect retrievers on the studied models in the LM task in Appendix D.1.

Moreover, Figure 9 illustrates the performance of the models on QA reasoning datasets. When evaluating REALM and kNN-LM on the StrategyQA dataset, we append *yes/no* to each statement. These results highlight the superiority of Contriever + Flan-T5 in our experiments that matches our finding in Section 4.3.2.

In order to study which component (retriever or LM) is more responsible for the failures of the retriever-augmented LMs, we make a hypothetical assumption. We assume that we have prior knowledge of the exact number of statements (k) to be retrieved for each input. We then report the number of Contriever + Flan-T5's failure examples (i.e., samples with F1 score less than 0.5) in Appendix D.3. Out of a total of 340 data sam-

ples, it is noteworthy that the retriever misses at least one gold statement in nearly 85% of the cases. Among these, only 19% are generated correctly by the LM. Conversely, when the retriever performs flawlessly (i.e., no missing gold statements), the LM correctly responds in 34% of the cases. In summary, retriever-augmented LMs' failures appear to be more attributable to the retriever. This is underscored by the noticeable improvement in the LM's performance (34% compared to 19%) when the retriever performs perfectly, emphasizing the pivotal role of the retriever in achieving correct generation.

### 4.3.4 The Impact of Model Size

This subsection examines the influence of model size on the models' performance in reasoning tasks. Specifically, we analyze the performance of different variations of Flan-T5 coupled with Contriever on the EB-Hard dataset.

The experimental results in Figure 10 demonstrate that larger models achieve better F1 score. However, there is still a large room for improvement as Flan-T5-xxl achieves only 51.8% F1 when

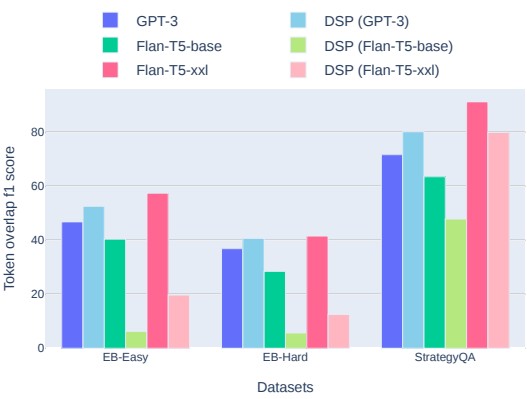

Figure 11: **Token overlap F1 score of GPT-3.5 and Flan-T5 variations using multihop DSP program.** All the experiments are done with few-shot examples using Contriever as the retriever and 5 retrieved statements in each retrieval step. The experimental results show that while DSP improves GPT-3.5 performance, it does not generalize to Flan-T5 models in the F1 score.

it has to reason over statements but the upperbound performance is 59.7% F1 when provided with the single oracle statement. The performance of the models on other QA datasets is presented in Appendix F.

#### 4.3.5 Impact of Multihop Retrieve-and-Read

To address the limitations of retrieve-then-read models on reasoning datasets, we explore the effectiveness of a strong multihop retrieve-and-read framework known as Demonstrate-Search-Predict (DSP), employing Contriever along with various language models (Khattab et al., 2022). The implementation details of the multihop retrieval experiments are provided in Appendix G.

The token overlap F1 scores of the models using the multihop DSP approach are depicted in Figure 11. The results indicate that while DSP improves GPT-3.5's performance, it still falls short compared to the retrieve-then-read Flan-T5-xxl. This observation shows a large room for improvement in multihop retrieve-and-read methods. Furthermore, we observe a decline in Flan-T5's performance when using multihop retrieval. This can be attributed to Flan-T5's inability to generate appropriate subqueries for the given question. Additionally, its generated responses tend to include all retrieved information, leading to high recall but low precision scores. This phenomenon is further exemplified in the qualitative examples and recall scores of the models, as demonstrated in Appendix G.

In summary, the results demonstrate the potential

of multihop retrieve-and-read for large language models like GPT-3.5, but it does not generalize to other models.

## 5 Conclusion

This paper analyzes the reasoning abilities of retriever-augmented language models. We first evaluate popular retrieve-then-read models, including $k$NN-LM, REALM, DPR + FiD, Contriever + ATLAS, and Contriever + Flan-T5, through language modeling and question answering tasks.

Our experimental results indicate that retrievers fail to select all essential statements for reasoning when relying on the similarity between the query and statements. Moreover, we observe that language models also struggle to reason over statements even when distracting information is absent. The performance deteriorates further when coupled with imperfect retrievers, as Flan-T5's performance drops by 28.6% when retrieving 5 statements using Contriever. Furthermore, while larger language models show greater capabilities in reasoning, they still have a large room for improvement. Additionally, our experiments on multihop retrieve-and-read show improvements on GPT-3.5, but these improvements do not generalize to other language models such as Flan-T5-xxl.

These findings present opportunities for future research to enhance the performance of retrieve-then-read models by addressing the aforementioned limitations of retrievers or language models. Moreover, the development of multihop retrieve-and-read models holds promise for advancing reasoning tasks.

## 6 Limitations

This paper examines the reasoning abilities of widely-used retriever-augmented language models in both LM and QA settings. To ensure a fair comparison, we employ models with similar sizes in both LM and QA tasks. Additionally, we investigate the impact of employing larger language models and a recent, strong multihop retrieve-and-read approach on the performance of these models.

In this paper, while one can finetune the models on the specific data, we focus on the capabilities of the already pretrained retriever-augmented language models. Moreover, we analyze the models in two basic NLP tasks, including language modeling and question answering. However, one of the studied models kNN-LM has been pretrained only for

language modeling exclusively, which may have resulted in a subpar performance on the QA task. Surprisingly, kNN-LM also performs the worst at the LM task compared to others.

Finally, unlike the popular retriever-based methods in the literature which use large corpus of knowledge, we use a data-specific set of statements for each data sample. This allowed us to have more control over the retrieved statements and the reasoning behavior of the language models.

## 7 Ethics Statements

This is not applicable to this work.

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

## A  Implementation Details

We present the implementation details of the analyzed models in this section. Most of the experiments are conducted using PyTorch (Paszke et al., 2019) on an RTX8000 GPU with 48GB memory in a single run, each taking a few minutes to run. We also run the experiments with large 30B-parameter models on an A100 GPU with 80GB memory. Note that we have changed the retriever in each model to retrieve statements from a sample-specific set of statements instead of a large common corpus.

In REALM's experiments, we use the Huggingface's transformers implementation for both masked language modeling and question answering (Wolf et al., 2020). We load the `realm-cc-news-pretrained-encoder` checkpoint as a knowledge encoder for masked language modeling and `realm-orqa-nq-openqa` checkpoint for question answering. For $k$NN-LM experiments, we use the best checkpoint available in the original papers' GitHub repository, and we find $\lambda = 0.65$ the best value as the interpolation hyperparameter based on the experiments on EntailmentBank development sets. In FiD experiments, we use `nq_reader_base` checkpoint available in the papers' GitHub repository with using the `nq.bert-base-encoder`'s checkpoint of the DPR retriever which is available in their GitHub repository. For experimenting ATLAS, we use the trained `atlas_data/models/atlas_nq/base` checkpoint of both the language model and retriever. Also, for the Flan-T5 model, we load the `flan-t5-base` model from Huggingface's transformers to be almost the same size as the other models in the main experiments.

In order to analyze the impact of the model size, we experiment with various Flan-T5 models from Huggingface. Our multihop retrieval experiments employ Contriever as addressed before with OpenAI's GPT-3.5 text-davinci-002 and different sizes of Flan-T5 model. More concretely, we use the same query formatting and templates as provided in the DSP GitHub repository.

## B  Dataset Preparation Details

In the language modeling experiments, we evaluate the models in target ranking task, where the model should assing a higher likelihood to the correct sentence compared to some alternative ones. Therefore, we first create an LM reasoning dataset for StrategyQA by changing the questions and

| Model | Target ranking score |
|---|---|
| REALM | $\log \frac{1}{|S_r|} \sum_{s_j \in S_r} p\left(\texttt{[MASK]} = T|Q, s_j\right) p\left(s_j|Q\right)$ |
| $k$NN-LM FiD Flan-T5 | $\frac{1}{N} \log p\left(Q_T|S_r\right)$, where $Q_T$ is the query $Q$ with $\texttt{[MASK]}$ tokens substituted with $T$ |
| ATLAS | $\frac{1}{M+1} \log p\left(\texttt{<extra\_id\_0>}\, T|Q, S_r\right)$ |

Table 5: **The target scoring function employed by each model.** Our language modeling task uses target ranking score to rank multiple sentences with different target candidates filled-in as answers. Target entity mention is indicated by $T = t_1 t_2 \ldots t_M$, input query by $Q = q_1 q_2 \ldots q_N$, and given retrieved statements by $S_r$. For ATLAS, we found that the T5 setup of predicting the mask with `<extra_id_0>` performed slightly better than computing the probability of the entire sentence.

the *yes/no* answers into declarative-form sentences. This is because StrategyQA samples only include questions as queries, not sentences. We also use hypothesis sentences of the EntailmentBank dataset for LM experiments. The next step includes creating alternative sentences for each sample for the target ranking task in our language modeling experiments. We keep the data samples that include at least one entity mention and mask out the last entity mention in the sentences of StrategyQA and EntailmentBank using Spacy (Honnibal and Montani, 2017). Also, we randomly pick at most four other entities mentioned in the data sample's statements as the alternative targets (as described in Section 3) and compare the model's score for each target. Regarding the question answering experiments, we use datasets' question and answer formats.

For the experiments on the EntailmentBank datasets, we run the experiments on the same development and test sets as the original data. However, in the StrategyQA dataset, since we do not have access to the answers in the test split, we cannot change the samples' formats to declarative form. Therefore, we pick 25% and almost 35% of the train data as the development and test sets, respectively.

## C  Model and Task-Specific Query Format

This section includes the model-specific query formats in each task. As stated in Section 3, we aim to study the reasoning abilities of retriever-augmented language models in language modeling and question answering tasks.

| Model | Alternative Target Scores |
|---|---|
| REALM | $\log \frac{1}{|S_r|} \sum_{s_j \in S_r} p\left([\text{MASK}] = \text{lithosphere}|Q, s_j\right) p\left(s_j|Q\right)$ 
 $\log \frac{1}{|S_r|} \sum_{s_j \in S_r} p\left([\text{MASK}] = \text{coal}|Q, s_j\right) p\left(s_j|Q\right)$ |
| $k$NN-LM 
 FiD 
 Flan-T5 | $\frac{1}{8} \log p(\text{Surface mining affects the lithosphere and biosphere.}|S_r)$ 
 $\frac{1}{8} \log p(\text{Surface mining affects the coal and biosphere.}|S_r)$ |
| ATLAS | $\frac{1}{2} \log p\left(\texttt{<extra\_id\_0>} \text{ lithosphere}|Q, S_r\right)$ 
 $\frac{1}{2} \log p\left(\texttt{<extra\_id\_0>} \text{ coal}|Q, S_r\right)$ |

Table 6: **A sample of the ranking strategies for each model for target ranking in the LM task using retrieved statements $S_r$.** The query ($Q$) in this example is "Surface mining affects the [MASK] and biosphere." with alternative targets "lithosphere" and "coal". For ATLAS experiments, we replace [MASK] with <extra_id_0> which is the specific masking token in T5-based models.

## C.1 Language Modeling

As explained in Section 3, we evaluate the performance of the popular retriever-augmented language models in the language modeling task through the *target ranking* problem. In this task, the model should assign a higher likelihood to the correct sentence compared to (up to) four alternative similar but incorrect sentences. These candidate sentences are generated by replacing the masking tokens with alternative entities available in the knowledge statements of the data sample. Table 5 depicts the target scoring functions employed by each model for computing the score of each candidate sentence. These alternative target scoring functions are further exemplified in Table 6 for a better understanding. The way models incorporate retrieved statements is explained in Section 2.3. In language modeling setup, the same happens for each model, except for REALM, which LM and QA variants differ. In QA, each statement is assigned a score separately, and the predicted span is the one with the highest retriever and reader score. In LM setup, instead, the score of each alternative target is computed by marginalizing over different retrieved statements, as shown in Table 5.

## C.2 Question Asnwering

In the question answering setting, on the other hand, we give the whole question to the model and take the generated output as the answer for Entailment-Bank datasets. In StrategyQA QA experiments, we compute how often models rank the correct *yes* or *no* answer higher than the other.

# D Main Quantitative Results

This section includes more visualizations and detailed results.

## D.1 Language Modeling

First, we report the overall performance of retriever-augmented language models in the reasoning LM task in Figure 12. It can be observed that Flan-T5, an instruction-tuned QA model, performs as the best model. On the other hand, $k$NN-LM, a pretrained language model, performs as the worst model in our LM task.

Additionally, we demonstrate the impact of additional distracting information in Figure 13. Results indicate that similar to QA experiments, in LM task, providing the language model with additional distracting information as well as all the required statements generally leads in performance degradation.

Furthermore, we illustrate the impact of the imperfect retrievers on language models in Figure 14. Results show that similar to the case with question answering task, the performance of the language models becomes even worse in LM when combined with imperfect retrievers.

Table 7 demonstrates the performance of the best retriever-augmented models (based on the performance on the dev sets) on the test sets in the language modeling task.

## D.2 Question Answering

We demonstrate the performance of the models on EB-Easy and EB-3 test set in question answering task in Figure 15. It can be observed that and Contriever + Flan-T5 performs the best in these datasets, which matches our findings in the main paper. We observe that the models' performance is similar to EB-Hard dataset with both relevant and distracting statements.

We report the performance of the best retriever-augmented models (based on the performance on the development sets) on the test sets in question answering in Table 8.

## D.3 Blame Game Between Retriever and Language Model

Our analysis of the Contriever + Flan-T5's responses in a hypothetical scenario (i.e., the exact number of retrieval is known) is provided in Table 9. Results show that the retriever does not retrieve all

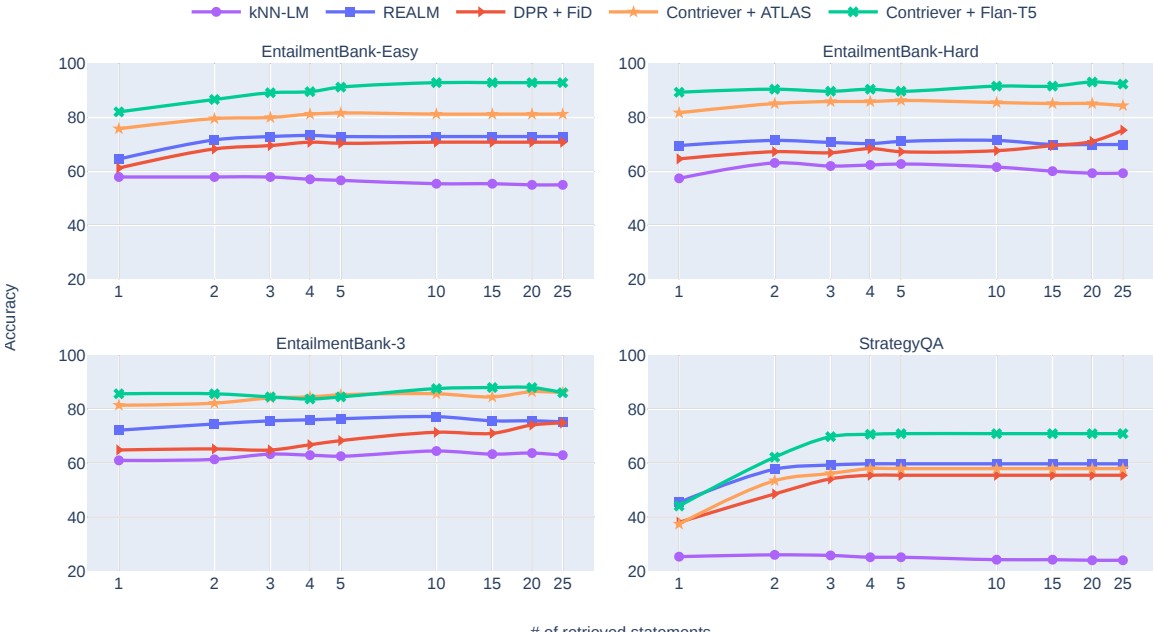

Figure 12: **The accuracy of the retriever-augmented language models in the target ranking (LM) problem.** The results show that Contriever + Flan-T5 and $k$NN-LM perform as the best and the worst models in our LM experiments.

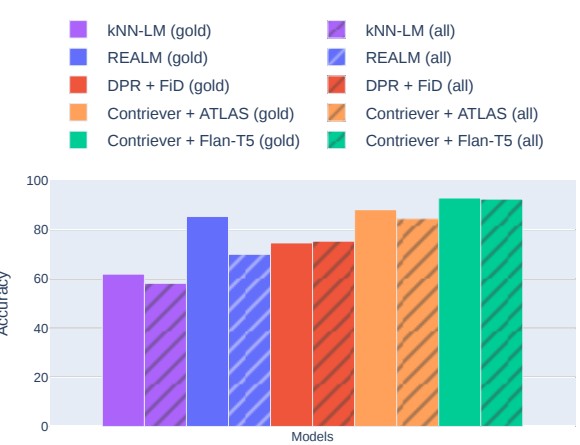

Figure 13: **The negative impact of additional distracting information on language models' performance in the LM task.** The solid bars and the bars with patterns refer to the experiments with all the gold statements and all gold and distracting statements, respectively. It can be observed that providing language models with distracting statements hurts their performance.

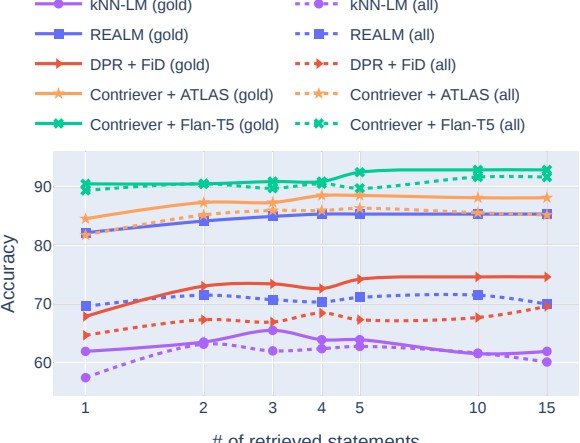

Figure 14: **Performance of language models on EB-Hard test set in the LM task coupled with imperfect retrievers.** Coupling language models with imperfect retrievers (i.e., retrieving some distracting statements as well as some required ones) deteriorates the overall performance.

the required statements most of the time. Additionally, when the retriever performs perfectly, the LM responds to the question correctly more often.

## E   Main Qualitative Results

In Section 4.3.2, we explained that F1 scores of the models given only the oracle statements are lower than expected. Table 10 demonstrates a few more

examples of the Flan-T5 failures.

We also demonstrate some failure examples in each of the retrievers and language models in Table 11 where imperfect retrievers and LMs are combined. In this table, a few true retrieved statements and the one that had to be retrieved in order for the model to solve the task correctly are highlighted in green and red, respectively. The true answer (or

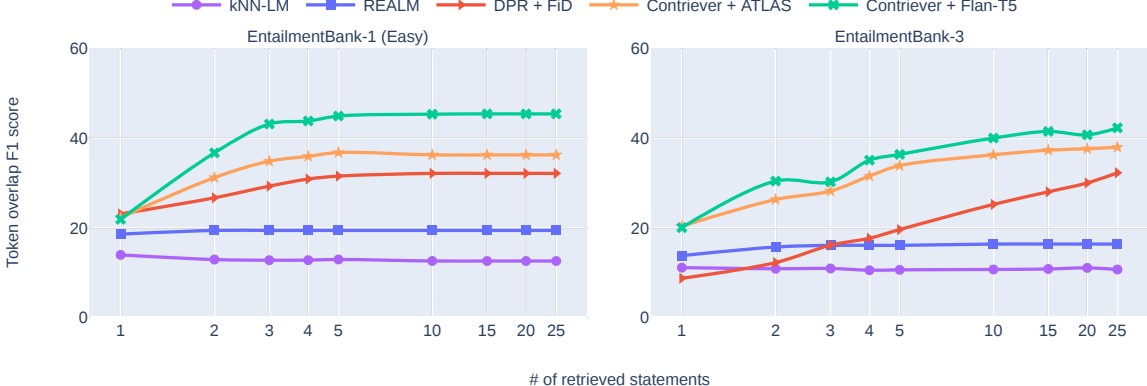

Figure 15: **The performance of the retriever-augmented language models in the QA task on EB-Easy and EB-3 datasets.** In EB-3, we observe that models perform similarly to EB-Hard including both gold and distracting supporting information. Furthermore, we observe that in EB-Easy models perform similarly to our experimental settings on EB-Hard with only gold statements, as EB-Easy consists of only required statements with no additional distracting information.

| | Language Modeling Accuracy | | | |
| | EB-Easy | EB-Hard | EB-3 | SQA |
|---|---|---|---|---|
| *k*NN-LM | 57.92 | 62.74 | 61.39 | 25.06 |
| REALM | 72.92 | 71.48 | 77.22 | 59.73 |
| DPR + FiD | 70.83 | 67.68 | 71.43 | 55.48 |
| Contriever + ATLAS | 81.25 | 85.55 | 84.56 | 57.94 |
| Contriever + Flan-T5 | **92.92** | **91.63** | **87.64** | **70.92** |

Table 7: **Experimental results of the best retriever-augmented models in LM on test sets.** The two best models are highlighted in green. The results show that Flan-T5 and ATLAS are superior in language modeling based on the target ranking accuracy.

| | Token overlap F1 score | | | Accuracy |
| | EB-Easy | EB-Hard | EB-3 | SQA |
|---|---|---|---|---|
| *k*NN-LM | 12.62 | 8.13 | 10.95 | 23.94 |
| REALM | 19.43 | 13.14 | 16.39 | 46.76 |
| DPR + FiD | 32.14 | 27.32 | 32.27 | 46.98 |
| Contriever + ATLAS | 35.95 | 33.14 | 37.63 | 53.69 |
| Contriever + Flan-T5 | **45.33** | **39.68** | **42.29** | **67.34** |

Table 8: **Experimental results of the best retriever-augmented models in QA on test sets.** The two best models are highlighted in green. The results show that Flan-T5 and ATLAS are the superior models in all of the studied datasets.

sequence of tokens leading to the true answer) for each data sample's statements is marked in bold. These examples explain how not retrieving the necessary statements for reasoning or not reasoning over true statements can lead to incorrect answers.

# F   Impact of the model size

In this paper, we compare various models from Flan-T5-small to Flan-T5-xxl with 80M to 11B parameters, respectively. The performance of these Flan-T5 models accompanied with Contriever on EB-Easy and StrategyQA QA datasets is presented in Figure 16. Experimental results show that larger models perform better in reasoning tasks.

| | Imperfect Retriever | Perfect Retriever |
|---|---|---|
| LM is Correct | 54 (19%) | 17 (34%) |
| LM is Incorrect | 236 (81%) | 33 (66%) |

Table 9: **An analysis on the blame game between the retriever and LM.** This table shows the number of failures of Contriever + Flan-T5 on EntailmentBank samples when the number of retrieved statements is known. Results highlight the noticeable improvement in the LM's performance (i.e., 34% compared to 19%) when the retriever operates perfectly. *Imperfect retriever* stands for cases where retriever misses at least one statement and *Incorrect LM* refers to cases where the prediction of the LM is marked as incorrect (i.e., samples with less than 0.5 F1 score).

| |
|---|
| *Model's response is semantically correct.* |
| **Question**: Many animals are still being hunted for their fur. Because of this, many of these animals are in danger of
**Statement**: If hunting decreases the animal population to zero, then the animal will be extinct.
**Expected answer**: extinction |
| **Response**: extinct |
| *Model's response is incorrect wrt the expected answer but is correct wrt commonsense.* |
| **Question**: Plants use energy directly from the sun. What do they use the energy from the sun for?
**Statement**: Plants use energy from the sun to make food.
**Expected answer**: to make food |
| **Response**: to grow |
| *Model's response is completely incorrect.* |
| **Question**: Which is a step in the process of photosynthesis?
**Statement**: Taking in carbon dioxide is a step in the photosynthesis process.
**Expected answer**: plants taking in carbon dioxide |
| **Response**: releasing light |

Table 10: **The predictions of Flan-T5 when the oracle statement is provided.** Statistics of the correct and incorrect responses are shown in Figure 6.

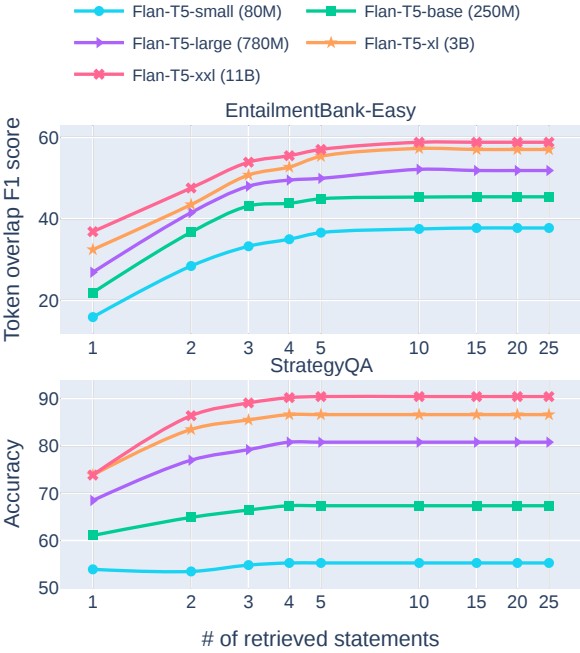

Figure 16: **Token overlap F1 score of various sizes of Flan-T5 on EB-Easy and StrategyQA test sets based on the number of retrieved statements.** The results demonstrate that larger models perform better in F1 scores. We use Contriever as the retriever in all experiments.

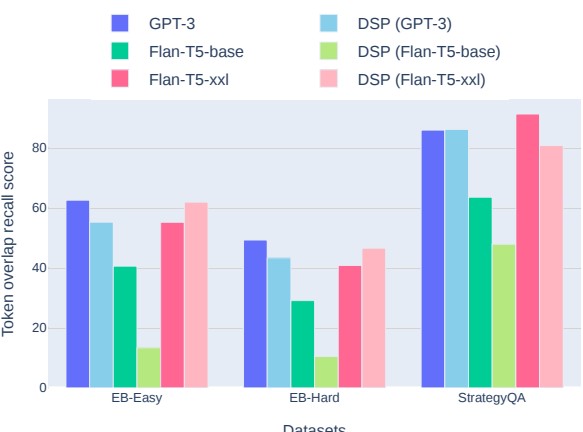

Figure 17: **The token overlap recall score of GPT-3.5 and Flan-T5 models using multihop DSP program.** The results show that Flan-T5-xxl achieves high recall score with multihop retrieval which can be due to including all the retrieved statements in the response. Although getting high recall score, this is not a desired behavior from language models.

## G   Impact of multihop retrieval

This section reports more details about the studied multihop retrieval approach (DSP). In multihop retrieval experiments, we retrieve 5 statements in each retrieval, with the same templates of the original DSP paper (Khattab et al., 2022). Due to the generally longer context size in multihop retrieval setting and the Flan-T5's context window limitations, multihop and retrieve-then-read experiments include two and five few-shot demonstrations in the prompt, respectively. Some of the qualitative examples of the multihop retrieval approach of DSP using GPT-3.5 and Flan-T5-xxl are illustrated in Table 12. Even though Flan-T5-xxl includes the correct answer tokens in its generated response, it can be observed that the subqueries generated by this model are sometimes nothing but paraphrasing or repetition of the original questions, while the goal of the multihop DSP program is to break down the problems into smaller subproblems. Moreover, the Flan-T5-xxl's responses usually include all the retrieved information, which is not desired. The recall score of the models using the multihop DSP approach is illustrated in Figure 17 which shows the relatively high recall score of the larger Flan-T5 model due to the problem mentioned above.

| | Model | Query | Statements | Answer |
|---|---|---|---|---|
| **Retriever's Failures** | DPR + Flan-T5 | In a zoo located in a warm region, what should be included in the polar bear exhibit? | + If an animal lives a certain environment then that animal usually requires that kind of environment. | a polar bear |
| | DPR + FiD | | - Polar bears live in **cold environments**. | warm |
| | Contriever + ATLAS | What keeps the Moon orbiting Earth? | - Moons orbit planets. 
 - **Gravity** causes orbits. | elliptical |
| | *k*NN-LM | The robot will weigh less on mars than earth but will have the same [MASK]. 
 Targets: *mass* vs *mars* | + As the force of gravity decreases, the weight of the object will decrease. 
 - The gravitational force of a planet does not change the **mass** of an object on that planet or celestial body. | mars |
| | REALM | A complete orbit of mercury around the sun takes [MASK]. 
 Targets: *around 88 earth days* vs *between 1 and 365* | + A complete revolution / orbit of a planet around its star takes 1 / one planetary year. 
 - One mercury year is **about 88 earth days**. | between 1 and 365 |
| | | If a new moon occurred on June 2, when will the next new moon occur? | + A new moon occurred on **june** 2. 
 + A moon phase occurs 28 days after the last time it occurred. 
 - 2 plus 28 equals **30**. | june 2 |
| **Language Model's Failures** | DPR + Flan-T5 | What allows two students standing ten feet apart to hear each other talk? | + Talking is when a human produces sound to communicate. 
 + Sound can travel through air by **vibrating air**. | a micro-phone |
| | DPR + FiD | Which energy conversion happens when a person shivers and the energy is transferred to make the muscles and joints move? | + A person is a kind of animal. 
 + When an animal moves, **chemical energy is converted to mechanical energy**. 
 + Shivering is a kind of shaking. 
 + Shaking is a kind of moving. | shaking |
| | Contriever + ATLAS | Wave energy from the ocean can be harnessed to power generators to make electricity. Energy from ocean tides can also be used to make electricity. How would you categorize these two sources of energy? | + Tidal energy means energy from ocean tides. 
 + Tidal energy is a **renewable resource**. 
 + Wave energy is a **renewable resource**. | Wave energy |
| | | Which changes will most likely have a negative effect on an ecosystem? | + Humans changing ecosystems usually has a negative impact on an ecosystem / organisms living in an ecosystem. 
 + **Humans building roads** in an ecosystem causes that ecosystem to change. | Humans changing ecosystems |
| | *k*NN-LM | The mass of earth causes the pull of gravity on [MASK]. 
 Targets: *earth* vs *newton* | + The mass of a planet causes the pull of gravity on **that planet**. 
 + **Earth** is a kind of planet. | newton |
| | REALM | Andy lives in the southern hemisphere. What season does he most likely experience in August? | + Andy lives in southern hemisphere. 
 + August is during the **winter** in the southern hemisphere. | in southern hemisphere |

Table 11: **A complete set of models' failure examples rooted in the retriever or language model modules.** In the retriever's failures section, a few correctly retrieved statements and the ones that had to be retrieved in order for the model to solve the task correctly are highlighted in green and red, respectively. The sequence of tokens leading to the true answer is marked in **bold**.

| Question | Statements | Answer |
|---|---|---|
| Which characteristic of a cheetah is more likely to be learned rather than inherited? | + Skills are learned characteristics.
+ Inherited characteristics are the opposite of learned characteristics.
+ Hunting is a kind of skill. | hunting strategies |

| GPT-3.5 | |
|---|---|
| first subquery | learned characteristics of a cheetah |
| first rationale | [1] and [2] suggest that skills are learned characteristics, while [3] suggests that hunting is a kind of skill. |
| second subquery | Is hunting a learned or inherited characteristic in cheetahs? |
| second rationale | First, we need to understand the definition of each term. [1] and [2] suggest that skills are learned characteristics, while [3] suggests that hunting is a kind of skill. Therefore, the answer must be a learned characteristic of a cheetah that is not hunting. |
| response | How to run |

| Flan-T5-xxl | |
|---|---|
| first subquery | What is the most likely characteristic of a cheetah to be learned rather than inherited? |
| first rationale | What is the most likely characteristic of a cheetah to be learned rather than inherited? |
| second subquery | What is the most likely characteristic of a cheetah to be learned rather than inherited? |
| second rationale | What is the most likely characteristic of a cheetah to be learned rather than inherited? |
| response | hunting is kind of skill inherited characteristics are opposite of learned characteristics answer hunting |

| Question | Statements | Answer |
|---|---|---|
| A bee depends on certain flowers for food. The flowers depend on the bee to | + Pollination is when pollinating animals, wind, or water carry pollen from one flower to another flower.
+ A bee is a pollinating animal.
+ Pollination requires pollinating animals.
+ Plant reproduction requires pollination. | carry pollen for repro-duction. |

| GPT-3.5 | |
|---|---|
| first subquery | What do bees eat? |
| first rationale | - A bee is a pollinating animal. - Pollination is when pollinating animals, wind, or water carry pollen from one flower to another flower. - Plant reproduction requires pollination. - Pollination requires pollinating animals. |
| second subquery | What is the relationship between bees and flowers? |
| second rationale | First, we need to know what pollination is. Pollination is when an animal carries pollen from one flower to another. This is important because flowers need to be pollinated in order to reproduce. Therefore, the bee depends on the flowers for food, and the flowers depend on the bee for pollination. |
| response | pollinate |

| Flan-T5-xxl | |
|---|---|
| first subquery | What do flowers depend on bees for? |
| first rationale | What do flowers depend on bees for? |
| second subquery | Pollination is when pollinating animals, wind, or water carry pollen from one flower to another flower. What do flowers depend on bees for? |
| second rationale | What do flowers depend on bees for? |
| response | Pollination is when pollinating animals, wind, or water carry pollen from one flower to another flower. |

| Question | Statements | Answer |
|---|---|---|
| A young boy and his brothers have green eyes. This trait is best described as | + Eye color is an inherited characteristic.
+ Green is a kind of color. | inherited |

| GPT-3.5 | |
|---|---|
| first subquery | What is a trait? |
| first rationale | - Eye color is an inherited characteristic. - Green is a kind of color. |
| second subquery | What is the definition of "inherited characteristic?" |
| second rationale | We are looking for a word that describes how the trait is passed down. This is an inherited characteristic, so it is passed down from parents to children. |
| response | inherited |

| Flan-T5-xxl | |
|---|---|
| first subquery | What is the medical term for the color of a person's eyes? |
| first rationale | What is the medical term for the color of a person's eyes? |
| second subquery | Eye color is an inherited characteristic. What is the medical term for the color of a person's eyes? |
| second rationale | What is the medical term for the color of a person's eyes? |
| response | Eye color is an inherited characteristic. Eye color is a kind of color. Eye color is inherited. The medical term for the color of a person's eyes is eye color. Eye color is an inherited characteristic. Eye color is a kind of color. |

Table 12: **Some examples of multihop question answering using the DSP approach with Contriever as the retriever.** In each sample, the generated subqueries, rationales, and final response are presented. The correct and incorrect answers included in the generated tokens are highlighted in green and red, respectively.