# OpenReview forum: "Can Retriever-Augmented Language Models Reason? The Blame Game Between the Retriever and the Language Model"
_EMNLP/2023/Conference — EMNLP 2023 Findings_

### Official Review · Reviewer_cXZR · 2023-08-01

**Soundness:** 4

**Excitement:**

4: Strong: This paper deepens the understanding of some phenomenon or lowers the barriers to an existing research direction.

**Missing References:**

The LM setup is very similar to the one in the COMPS and Surface Form competition papers, and these might be worth citing:

**Surface form competition:** https://aclanthology.org/2021.emnlp-main.564/

**COMPS:** https://aclanthology.org/2023.eacl-main.213/

In fact the COMPS paper also contains distracting knowledge statements, so it might also be relevant there.


**Paper Topic And Main Contributions:**

This paper presents a comprehensive evaluation of reasoning in retriever augmented language models (RA-LMs) -- language models with a separate module to retrieve statements that are relevant and important to solve the reasoning problem. To this end, the authors use existing reasoning datasets that require multiple steps of reasoning (in the form of statements)---EntailmentBank and StrategyQA---and augment them with additional statements that operate as distraction. This way, they are able to perform controlled evaluation of both the retriever and the language modeling component of the RA-LM, and characterize their strengths and weaknesses. Findings from a number of experiments suggests that: (1) similarity-based retrieval mechanisms are fundamentally limited in selecting statements that are useful for precise reasoning; (2) LMs seem to struggle even when they are provided with noise-free, ground truth statements that are sufficient to solve the reasoning problem; (3) there is an interaction effect between the LM and the retriever component, where their overall performance---expectedly---drops when the retrieved statements are insufficient for reasoning. Finally, the authors study a method involving multi-hop retrieval in addition to the standard retrieval process, and find that that while it augments the performance of off-the-shelf non-retriever based LMs like GPT3.5, it is unable to generalize to relatively smaller models such as FLAN-T5-XXL. Overall, findings from this work can be used to develop improved RA-LMs.

**Questions For The Authors:**

A: Do you have a conjecture/hypothesis as to why LMs do not show perfect behavior when they have been provided with the oracle statement in context? I think adding this in the paper will strengthen it.

**Reasons To Accept:**

- Controlled evaluation of retriever augmented LMs seems to be an understudied topic, and this paper does it quite well.
- Experiments are thorough and comprehensive, and the results sufficiently support the conclusions.
- The most exciting finding to me are both the fact that RA-LMs are unable to reason well even with ground truth statements, as well as the fact that the simple similarity based metric employed by retrievers is not well suited for picking out statements that aid in reasoning.


**Reasons To Reject:**

I do not see any major reasons to reject this paper.

However, I do want to highlight the fact that the main text of the paper is not completely sufficient to evaluate it. The method for creating statements with distraction is mentioned in the appendix, but is actually a critical component of the paper---it dictates how the retrievers will behave and has a causal implication for the results. Additionally, the way this method has been presented in the appendix is very condensed and lacks sufficient examples. I would strongly encourage the authors to consider this feedback, move the method to the main text, and present a proper characterization of the distraction statements. Furthermore, I found the description of the multi-hop retrieval to be insufficient--i.e., I am still unclear about the precise difference between this method and the setting where ground truth statements are provided to the model -- wouldn’t this only affect performance on questions that require multi-hop decomposition?

**Reproducibility:**

4: Could mostly reproduce the results, but there may be some variation because of sample variance or minor variations in their interpretation of the protocol or method.

**Reviewer Confidence:**

4: Quite sure. I tried to check the important points carefully. It's unlikely, though conceivable, that I missed something that should affect my ratings.

**Typos Grammar Style And Presentation Improvements:**

Line 061: “combining the details” seems a bit vague -- I encourage you to rephrase this.

I would recommend moving the “question” and “statement” parts above the Model and Answer part of the table in Table 3.

Line 463: replace overlapping → overlap

I would recommend converting Fig 6 into a paired line plot that can make the drop in performance when distraction is present more obvious.

This does not really impact my scores but the limitations do not read like limitations and instead as an additional contribution and discussion section of sorts.

---

> ### Author Rebuttal · Authors · 2023-08-28
>
> We would like to thank you for your comprehensive review and are grateful for your constructive comments. We are thrilled that you believe our paper provides:
>
> * **controlled evaluation** on an **understudied topic quite well**,
> * **thorough and comprehensive with results backing our claims**, as also emphasized by Reviewers WssQ and k18u,
> * **insightful findings**, as also highlighted by Reviewer WssQ.
>
> We are also delighted that other reviewers found the paper to be:
>  * well-written (Reviewers WssQ and k18u),
>  * well-organized (Reviewer WssQ),
>  * systematic (Reviewer k18u) and concrete (Reviewer WssQ) analysis of the recent (Reviewers WssQ and orzP), well-known (Reviewer WssQ), and representative (Reviewer orzP) models.
>
> We engage with your questions and suggestions below:
>
> > Move dataset preparation from the Appendix to the main text
>
> Thank you for pointing this out. We agree this is important and will move this to the main text in the camera-ready. We will also elaborate on the nature of distracting statements with examples. Most commonly, the distracting statements contain entities or relations mentioned in the input query which might make them deemed relevant but in actuality they are irrelevant.
>
> > Precise difference between the multi-hop method and the setting where ground-truth statements are provided to the model
>
> In the multi-hop setting, the model has two advantages: 1) it can decompose the problem into smaller problems, allowing it to generate intermediate hypothesis statements that might be useful in the future; and 2) retrieves only relevant facts for each subproblem and incrementally build the answer using retrieved facts and intermediate hypotheses. Previous studies [1] show that intermediate steps make complex reasoning easier. Whereas in the ground-truth setting, the model has access to the gold statements all at once and then has to generate an answer. We will elaborate on the differences and advantages in the camera-ready.
>
> > Reasons LMs do not show perfect behavior with oracle statement
>
> Thank you for raising this interesting question. We conducted **additional human analysis on 50 randomly sampled responses of Flan-T5** given the oracle statement (table below). We will add this analysis to the camera-ready since we believe it will be useful for the community and will strengthen our claims. As stated in Sec 4.3.2, upper-bound performance of the models is lower than expected because 1) "F1 evaluation metric penalizes when there is a word mismatch between true and predicted answer even if the predicted answer is semantically the same" (case 2 and case 3) which is also observed in [2, 3], and 2) sometimes a model like "Flan-T5 uses its internal knowledge rather than the provided statement" (case 6 and case 7).
>
> |Case|Oracle statement is relevant to the answer|Response is grounded in the oracle statement|Correctness of the response|Proportion|
> |---|---|---|---|---|
> |1|+|+| completely (full F1)| 34% |
> |2|+|+| less or more elaborate response | 22% |
> |3|+|+| semantically correct | 10% |
> |4|+|+| incorrect wrt answer but correct wrt general knowledge | 2% |
> |5|+|+| incorrect | 10% |
> |6|+|-| incorrect wrt answer but correct wrt general knowledge | 10% |
> |7|+|-| incorrect | 8% |
> |8|-|-| incorrect | 4% |
>
> We provide examples for case 3 and case 6.
> || Case 3 | Case 6 |
> |--- |---|---|
> |Question | Many animals are still being hunted for their fur. Because of this, many of these animals are in danger of | Plants use energy directly from the sun. What do they use the energy from the sun for? |
> |Oracle statement|If hunting decreases the animal population to zero, then the animal will be extinct.|Plants use energy from the sun to make food.|
> |Answer|extinction|to make food|
> |Response|extinct|to grow|
>
>
> > Missing References
>
> Thank you for pointing out these papers, we will discuss them in the camera-ready.
>
> We thank you again for your time and the positive assessment of our work. Given your assessment and these improvements, would you consider championing our paper for publication at this conference? We are happy to engage further.
>
> ### References
> [1] Khattab et al, DEMONSTRATE–SEARCH–PREDICT: Composing retrieval and language models for knowledge-intensive NLP, Arxiv 2023
>
> [2] Kamalloo et al, Evaluating Open-Domain Question Answering in the Era of Large Language Models, ACL 2023
>
> [3] Adlakha et al, Evaluating Correctness and Faithfulness of Instruction-Following Models for Question Answering, Arxiv 2023

---

### Official Review · Reviewer_k18u · 2023-08-03

**Soundness:** 4

**Excitement:**

3: Ambivalent: It has merits (e.g., it reports state-of-the-art results, the idea is nice), but there are key weaknesses (e.g., it describes incremental work), and it can significantly benefit from another round of revision. However, I won't object to accepting it if my co-reviewers champion it.

**Paper Topic And Main Contributions:**

This manuscript provides a comprehensive analysis of the limitations of retriever-augmented language models for reasoning over retrieved information from both the retriever and language model perspectives. The authors demonstrate that the failure of retriever-augmented language models to reason effectively can be attributed to either the retriever's inability to find the necessary statements or the language model's difficulty in reasoning over them.

**Reasons To Accept:**

The paper is well-written, and the experiments are conducted systematically.

**Reasons To Reject:**

[1] While the manuscript is well-written and the experiments are conducted systematically, the novelty of the findings may not be as significant as expected. It appears somewhat evident that the limitations of retriever-augmented language models stem from issues with either the retriever or the language models.
[2] Additionally, the authors employ a multi-hop retrieve-and-read approach to address the limitations of retrieve-and-read, which appears similar to the Demonstrate-Search-Predict (DSP) method without introducing novel concepts to overcome the limitations.

**Reproducibility:**

4: Could mostly reproduce the results, but there may be some variation because of sample variance or minor variations in their interpretation of the protocol or method.

**Reviewer Confidence:**

4: Quite sure. I tried to check the important points carefully. It's unlikely, though conceivable, that I missed something that should affect my ratings.

**Typos Grammar Style And Presentation Improvements:**

Regarding Table 3, I noticed that the expected answer 'gravitational' is not shown in the oracle statement.

---

> ### Author Rebuttal · Authors · 2023-08-28
>
> We appreciate your feedback regarding your concerns. We are happy that you found our paper to be:
> - **well-written**, as also highlighted by Reviewer WssQ,
> - a **comprehensive analysis**,
> - with **systematic experiments**.
>
> We are also thrilled that other reviewers found our paper:
> - well-organized (Reviewer WssQ),
> - with an interesting idea (Reviewer orzP),
> - comprehensive (Reviewer WssQ and cXZR), concrete (Reviewer WssQ), and controlled (Reviewer cXZR) analysis of the recent (Reviewer WssQ and orzP), well-known (Reviewer WssQ), and representative (Reviewer orzP) models,
> - with exciting (Reviewer cXZR) and insightful (Reviewer WssQ) findings.
>
> We respectfully disagree with the reviewer's criticisms of our work and would encourage them to read the paper carefully and other reviewers' comments.
>
> > Lack of novel findings... evident that the limitations of RALMs stem from either the retriever or the LM
>
> Certainly, we acknowledge that the failures of RALMs are indeed attributed to either of the two modules. However, we are the first to systematically study the shortcomings of several retrievers and LMs in similar experimental settings. We believe such systematicity is highly valuable in the current landscape of NLP where it is becoming hard to disentangle true strengths and weaknesses from confounding factors.
>
> Moreover, our paper had five insights (lines 79–94), a substantial contribution with insightful findings, all in one place.
>
> > No novel concept to overcome the limitations of the multi-hop retrieve-and-read approach, which appears similar to the Demonstrate-Search-Predict (DSP)
>
> Our work precisely states that we use DSP in our experiments. In line 456, we say that “we explore the effectiveness of a strong multi-hop retrieve-and-read framework known as Demonstrate-Search-Predict (DSP)”, not any other similar approach. Through our analysis, we examine the shortcomings of DSP on LLMs and demonstrate that this method heavily relies on the LM's ability to generate subqueries. Surprisingly, even large models like Flan-T5-xxl struggle in this aspect.
>
> Additionally, we highlight that while DSP enhances GPT-3.5's performance, it still falls behind the retrieve-then-read Flan-T5-xxl. This finding highlights the ongoing need for improvement in multi-hop retrieval methods.
>
> > Typo about ‘gravitational’ in Table 3
>
> No, this is not a typo. We again encourage you to read the paper attentively. We intentionally highlight this example to demonstrate a scenario where the F1 score penalizes the models' correct response 'gravity.'
>
> We kindly request you to reassess the work carefully in light of the clarifications we've provided.

---

### Official Review · Reviewer_orzP · 2023-08-05

**Soundness:** 4

**Excitement:**

3: Ambivalent: It has merits (e.g., it reports state-of-the-art results, the idea is nice), but there are key weaknesses (e.g., it describes incremental work), and it can significantly benefit from another round of revision. However, I won't object to accepting it if my co-reviewers champion it.

**Paper Topic And Main Contributions:**

This paper examines a few retrieval augmented language models' performance on reasoning intensive tasks: EntailmentBank and StrategyQA. The performance of both the retrievers and the readers are examined. 5 popular models are evaluated without fine-tuning on the EntailmentBank and StrategyQA datasets: kNN-LM, REALM, DPR + FiD, Contriever + ATLAS and Contriever + Flan-T5.

The main takeaways from the experiments are: (1) The retrievers of these models tend to fail to perfectly retrieve the necessary statements for the reasoning intensive tasks. (2) Even with the gold evidence, the neural readers could not perfectly answer the question. (3) When multi-step reasoning over the statements is needed to come up with the answer, different models perform differently (e.g., Flan-T5 has a better reasoning ability compared with others). (4) Bad retrieval quality can largely affect QA model's performance. (5) Large model usually has a better QA performance. (6) Multi-step retrieval is promising when combined with large language models.

**Reasons To Accept:**

 - The idea of evaluating the retriever and reader of the language models on reasoning intensive tasks is interesting.
 - The authors evaluate 5 recent and representative retrieval augmented neural models, which is informative to other readers.

**Reasons To Reject:**

 - One major drawback is that all of the models are only evaluated without any fine-tuning. Although it is good to know the zero-shot performance, knowing their limitation after fine-tuning would be much more helpful.
 - Some takeaways are already discussed in previous literatures, and the results are reported for similar models, which makes this work less exciting. E.g.,
   * For takeaways (2) and (4) above, similar results are reported in "Better retrieval may not lead to better question answering".
   * For takeaway (5) above, "Chain-of-Thought Prompting Elicits Reasoning in Large Language Models"

**Reproducibility:**

4: Could mostly reproduce the results, but there may be some variation because of sample variance or minor variations in their interpretation of the protocol or method.

**Reviewer Confidence:**

3: Pretty sure, but there's a chance I missed something. Although I have a good feel for this area in general, I did not carefully check the paper's details, e.g., the math, experimental design, or novelty.

---

> ### Author Rebuttal · Authors · 2023-08-28
>
> We would like to thank you for the positive and valuable feedback you have provided.  We are pleased that you find:
> - the idea of this paper to be **interesting**.
> - all the studied retriever-augmented language models (RALMs) to be **recent and representative**.
> - The findings to be **informative for other readers**.
>
> We are also delighted that other reviewers found the paper:
> - well-written (Reviewers WssQ and k18u),
> - well-organized (Reviewer WssQ),
> - a systematic (Reviewer k18u), comprehensive (Reviewers WssQ, k18u, and cXZR), concrete (Reviewer WssQ), and controlled (Reviewer cXZR) analysis of the recent and well-known models (Reviewer WssQ),
> - providing exciting (Reviewer cXZR) and insightful (Reviewer WssQ) findings.
>
> We engage with your concerns below:
>
> > Lack of finetuned experiments
>
> We believe that the community is shifting towards using language models 'as is', in zero-shot or few-shot settings, without further finetuning. Thus, as indicated in the limitations section, our focus is on evaluating the performance of popular retrievers and LMs on datasets they have not encountered during training. To ensure that we do not underestimate the models due to the absence of fine-tuning, we not only compare models with each other but also evaluate each model against its **upper-bound performance** when provided with an oracle statement.
>
> > Some already discussed takeaways
>
> We appreciate your reference to the papers. We find paper [1] intriguing and will include it in the final version. However, in contrast to [1], we are the first to systematically study the shortcomings of several retrievers and LMs in similar experimental settings. We believe such systematicity is highly valuable in the current landscape of NLP where it is becoming hard to disentangle true strengths and weaknesses from confounding factors. While previous work in RALMs focused on multihop QA datasets like HotpotQA, we focus on a controlled setting using EntailmentBank where the knowledge statements and queries may not have surface level lexical similarities. In our experiments, reasoning connections are not readily obvious from the questions, distinguishing it from previous work. Here is an example (edited for simplicity)
>
> || EntailmentBank | HotpotQA [2] |
> |---|---|---|
> | Question| What is Phobos? | The football manager who recruited David Beckham managed Manchester United during what timeframe? |
> |Gold knowledge| (a) Phobos orbits Mars, (b) Mars is a planet, (c\) Moons orbit planets | (a) United manager Sir Alex Ferguson... Beckham was part of a group of young talents Ferguson brought into United in the 1990s.., (b) .. best known for managing Manchester United from 1986 to 2013.|
>
> Here, the question in EntailmentBank has no lexical similarities with required statements (b) and (c\), and without a strong reasoning component, models would struggle to retrieve and reason upon such statements.
>
> While [3] explores model scale across various model types and datasets with chain-of-though prompting,  this paper investigates the impact of model size on performance in a zero-shot setup with distracting information. Our analysis shows while larger models outperform the smaller ones, they still fall short of their upper-bound performance. Surprisingly, our analysis highlights that despite GPT-3.5 being over 10 times larger than Flan-T5-xxl, it fails to achieve a superior token overlap F1 score, whether with or without DSP.
>
> We thank you for your constructive feedback. In addition to the above clarifications, we also intend to provide additional error analysis for the oracle experiment (upper-bound) and error attribution during the combination (please see response to Reviewer WssQ). Given these revisions, could you reconsider your evaluation of the excitement of our work?
>
>
> ### References
> [1] Liang et al, Better retrieval may not lead to better question answering, Arxiv 2022
>
> [2] Yang et al, HotpotQA: A Dataset for Diverse, Explainable Multi-hop Question Answering, EMNLP 2018
>
> [3] Wei et al, Chain-of-Thought Prompting Elicits Reasoning in Large Language Models, NeurIPS 2022

---

### Official Review · Reviewer_WssQ · 2023-08-05

**Soundness:** 4

**Excitement:**

4: Strong: This paper deepens the understanding of some phenomenon or lowers the barriers to an existing research direction.

**Paper Topic And Main Contributions:**

This paper presents a comprehensive analysis of recent approaches for augmenting pre-trained language models with retrievers such as kNN-LM, REALM, DPR + FiD, Contriever + ATLAS, and FLAN-T5. The core investigations highlight three main findings: (1) the existing similarity scores are deemed limited in retrieving adequate evidence for reasoning, (2) language models often fail to perform reasoning even with the golden evidence, and (3) with imperfect retrievers, their performances are degraded. Additionally, the paper includes an extended analysis of multihop retrievers using different language models.

**Questions For The Authors:**

- What would be the most interesting findings from Section 4.3.3 and 4.3.5?

- Is it possible to distinguish the retrieval error from the LM error? and how?

- (Minor) In Table 3, the authors could consider a small-scale case study with human annotators to strengthen the claim.

**Reasons To Accept:**

1. The paper is well-written and -organized, supporting their claims with proper experimental results.

2. Recent issues in combining retrieval systems with language models are clearly demonstrated in the concrete experimental setup.

3. The authors thoroughly analyze the issues in each component and provide insightful findings with their examples.

**Reasons To Reject:**

1. **Lack of novel findings**
While the paper sheds light on the choice of existing well-known frameworks and offers valuable insights into augmenting pre-trained language models with retrievers, it falls short in terms of novel findings. It reconfirms the well-known issues in each component, the failure in retrieval or reasoning over multiple statements. It could have provided novel insights in sections for combining two components or multi-hop QA; however not observed.

2. **Concerns about claims on "Blame Game"**
Usage of the word "*Blame Game*" in the title could mislead the readers. It implicates the questions about who bears more responsibility between the retriever and language model, which is not presented in the paper. For claiming it, the quantitative analysis of retrieval and LM error or the presence of unique issues in combining them should be provided.

**Reproducibility:**

4: Could mostly reproduce the results, but there may be some variation because of sample variance or minor variations in their interpretation of the protocol or method.

**Reviewer Confidence:**

4: Quite sure. I tried to check the important points carefully. It's unlikely, though conceivable, that I missed something that should affect my ratings.

**Typos Grammar Style And Presentation Improvements:**

- In line 855, ‘findings’ was misspelled as ‘fndings’.

---

> ### Author Rebuttal · Authors · 2023-08-28
>
> We would like to thank you for your positive and thoughtful feedback, as well as your constructive concerns and questions. We are delighted that you found our paper:
> - **well-written** (also highlighted by Reviewer k18u) and **well-organized**,
> - providing **insightful findings** with examples,
> - **clearly demonstrating recent issues** on well-known frameworks of retriever-augmented language models (RALM),
> - a **concrete experimental setup** and **comprehensive analysis**, as also mentioned by Reviewers k18u and cXZR.
>
> We are also delighted that other reviewers found the paper to be:
> - an interesting idea (Reviewer orzP),
> - a systematic (Reviewer k18u) and controlled (Reviewer cXZR) analysis of the recent and representative models (Reviewer orzP),
> - providing exciting findings (Reviewer cXZR).
>
> We address your concerns and questions below:
>
> > Lack of novel insights
>
> In contrast to the literature [1], we are the first to systematically study the shortcomings of several retrievers and LMs in similar experimental settings. We believe such systematicity is highly valuable in the current landscape of NLP where it is becoming hard to disentangle true strengths and weaknesses from confounding factors. While previous work on RALMs focused on QA datasets like Natural Questions and multihop like HotpotQA, we focus on a controlled setting using EntailmentBank where the knowledge statements and queries may not have surface-level lexical similarities. The reasoning connections are not readily obvious from the questions, distinguishing it from previous work. Here is an example (edited for simplicity)
>
> || EntailmentBank | HotpotQA [2] |
> |---|---|---|
> | Question| What is Phobos? | The football manager who recruited David Beckham managed Manchester United during what timeframe? |
> |Gold knowledge| (a) Phobos orbits Mars, (b) Mars is a planet, (c\) Moons orbit planets | (a) United manager Sir Alex Ferguson... Beckham was part of a group of young talents Ferguson brought into United in the 1990s.., (b) .. best known for managing Manchester United from 1986 to 2013.|
>
> Here, the question in EntailmentBank has no lexical similarities with required statements (b) and (c\), and without a strong reasoning component, models would struggle to retrieve and reason upon such statements.
>
> Moreover, our paper had five insights (lines 79--94). Even if some of them are reconfirmations of existing insights, we believe this is a substantial contribution, all in one place. As you pointed in the strengths, these are *`insightful findings`*. Would you reconsider your comment about lack of novel insights in the reasons for rejection? We will add more discussion on the novelty in the paper.
>
> > "Concerns about overclaims on Blame Game” and "who bears more responsibility between the retriever and language model" (also Question 2)
>
> We thank you for raising this important question. We have provided scenarios that we thought were critical, i.e., retrieval-only failures (Sec 4.3.1), LM-only failures (Sec 4.3.2), and the realistic combination of them (Sec 4.3.3). If you mean pin-pointing precisely whose failure it is for a given example, yes, we provide additional results but some assumptions could be unrealistic.
>
> Assuming we know the exact number of statements (*k*) to be retrieved for a given input, we report the number of results with Contriever as retriever and Flan-T5 as LM (where samples with F1 score less than 0.5 are considered incorrect):
>
> |  |retrieval missing a statement | all required statements retrieved |
> | --- |---|---|
> | **LM is incorrect** | 236 | 33 |
> | **LM is correct** | 54 | 17 |
>
> However, assuming the exact number of statements *k* to be already known is a strong assumption. If we relax this assumption, then it becomes a complex problem, which is why in the combination section we provided all the ranges of 'k' (Fig 7).
>
> In any case, you raise an interesting point. We are happy to include the above table for different RALMs if it is useful. Given this, would this change your mind about overclaim? Thank you!
>
>
> > Interesting findings from Sec 4.3.3 and 4.3.5 (Question 1)
>
> By comparing the outcomes of Sec 4.3.2 (Fig 5) and Sec 4.3.3 (Fig 7), we illustrate the contributions of LMs and retrievers to the overall shortcomings. Moreover, Fig 7 uncovers additional interesting observations. Notably, the larger gap between the solid and dotted lines for Flan-T5, in contrast to Atlas, demonstrates that providing gold statements to Flan-T5 yields a more significant performance improvement (despite both using the same retriever and retrieved statements). The impact of a subpar retriever is also evident. While FiD and Atlas perform nearly equally well with gold statements, the performance of DPR+FiD notably lags behind Contriever+Atlas when given all (distracting + gold) statements.
>
> Previous studies [3] show that multi-hop retrieve-and-read makes complex reasoning easier. In Sec 4.3.5, Fig 10 illustrates that DSP detrimentally affects Flan-T5's performance but enhances that of GPT-3. However, even with the improved F1 score for GPT-3 (light blue bar), it still falls short of the retrieve-then-read approach of Flan-T5-xxl (dark pink bar), which underscores the need for further improvement in multi-hop retrieve-and-read techniques.
>
> > A small-scale case study for Table 3 (Question 3)
>
> We appreciate your constructive comment. Our **further analysis on 50 randomly sampled Flan-T5 responses** shows that 18% of the Flan-T5 responses are not grounded in the oracle statement, even though the oracle statement was relevant to the answer (case 6 and case 7). Additionally, 32% of the responses yield a low F1 score even though they are correct (case 2 and case 3, as also highlighted in [4, 5]). The detailed statistics from our analysis (table below) will be added to the paper to provide clarity to the community.
>
> |Case|Oracle statement is relevant to the answer|Response is grounded in the oracle statement|Correctness of the response|Proportion|
> |---|---|---|---|---|
> |1|+|+| completely (full F1)| 34% |
> |2|+|+| less or more elaborate response | 22% |
> |3|+|+| semantically correct | 10% |
> |4|+|+| incorrect wrt answer but correct wrt general knowledge | 2% |
> |5|+|+| incorrect | 10% |
> |6|+|-| incorrect wrt answer but correct wrt general knowledge | 10% |
> |7|+|-| incorrect | 8% |
> |8|-|-| incorrect | 4% |
>
> We provide examples for cases 3, 6, and 7 in the following table:
>
> || Case 3 | Case 6 | Case 7 |
> |--- |---|---|---|
> |Question | Many animals are still being hunted for their fur. Because of this, many of these animals are in danger of | Plants use energy directly from the sun. What do they use the energy from the sun for? | Which is a step in the process of photosynthesis? |
> |Oracle statement|If hunting decreases the animal population to zero, then the animal will be extinct.|Plants use energy from the sun to make food.|Taking in carbon dioxide is a step in the photosynthesis process.|
> |Answer|extinction|to make food|plants taking in carbon dioxide|
> |Response|extinct|to grow|releasing light|
>
> We again thank you for your insightful comments which will improve the paper further. We intend to incorporate them into the paper. With these revisions, could you reassess your excitement of our work?
>
> ### References
> [1] Liang et al, Better retrieval may not lead to better question answering, Arxiv 2022
>
> [2] Yang et al, HotpotQA: A Dataset for Diverse, Explainable Multi-hop Question Answering, EMNLP 2018
>
> [3] Khattab et al, DEMONSTRATE–SEARCH–PREDICT: Composing retrieval and language models for knowledge-intensive NLP, Arxiv 2023
>
> [4] Kamalloo et al, Evaluating Open-Domain Question Answering in the Era of Large Language Models, ACL 2023
>
> [5] Adlakha et al, Evaluating Correctness and Faithfulness of Instruction-Following Models for Question Answering, Arxiv 2023

---

### Meta-Review · Area_Chair_BXqY · 2023-09-19

**Recommendation:** 3

**Metareview:**

This paper presents a comprehensive analysis of recent approaches for augmenting pre-trained language models with retrievers (kNN-LM, REALM, DPR + FiD, Contriever + ATLAS, and FLAN-T5) and why they fail in reasoning (EntailmentBank and StrategyQA). The main findings are:
1. the existing similarity scores are inadequate  in retrieving adequate evidence for reasoning,
2. LLMs often fail to perform reasoning even with the golden evidence, and
3. with imperfect retrievers, their performances are degraded.
4. Larger LLMs have  better QA and Multi-step retrieval performances.

Strength:
1. The paper is well-written and well-executed.
2. The authors thoroughly analyze the issues in each component and provide insightful findings with their examples.

Weakness:
1. Lack of novel findings
2. all of the models are only evaluated without any fine-tuning. Although it is good to know the zero-shot performance, knowing their limitation after fine-tuning would be much more helpful.

---

### Decision · Program_Chairs · 2023-10-07

**Decision:**

Accept-Findings

**Comment:**

This paper presents a comprehensive analysis of recent approaches for augmenting pre-trained language models with retrievers (kNN-LM, REALM, DPR + FiD, Contriever + ATLAS, and FLAN-T5) and why they fail in reasoning (EntailmentBank and StrategyQA). The main findings are:
1. the existing similarity scores are inadequate  in retrieving adequate evidence for reasoning,
2. LLMs often fail to perform reasoning even with the golden evidence, and
3. with imperfect retrievers, their performances are degraded.
4. Larger LLMs have  better QA and Multi-step retrieval performances.

Strength:
1. The paper is well-written and well-executed.
2. The authors thoroughly analyze the issues in each component and provide insightful findings with their examples.

Weakness:
1. Lack of novel findings
2. all of the models are only evaluated without any fine-tuning. Although it is good to know the zero-shot performance, knowing their limitation after fine-tuning would be much more helpful.